# Development of New Strategies for Malaria Chemoprophylaxis: From Monoclonal Antibodies to Long-Acting Injectable Drugs

**DOI:** 10.3390/tropicalmed7040058

**Published:** 2022-04-07

**Authors:** Joerg J. Moehrle

**Affiliations:** Integrated Sciences, R&D, Medicines for Malaria Venture, Route de Pré Bois 20, CH-1215 Geneva 15, Switzerland; moehrlej@mmv.org

**Keywords:** malaria, *P. falciparum*, *P. vivax*, chemotherapy, chemoprevention, intermittent preventive treatment, mass drug administration, antimalarial, monoclonal antibodies, seasonal malaria chemoprevention, prophylaxis, chemoprophylaxis

## Abstract

Drug discovery for malaria has traditionally focused on orally available drugs that kill the abundant, parasitic blood stage. Recently, there has also been an interest in injectable medicines, in the form of monoclonal antibodies (mAbs) with long-lasting plasma half-lives or long-lasting depot formulations of small molecules. These could act as prophylactic drugs, targeting the sporozoites and other earlier parasitic stages in the liver, when the parasites are less numerous, or as another intervention strategy targeting the formation of infectious gametocytes. Generally speaking, the development of mAbs is less risky (costly) than small-molecule drugs, and they have an excellent safety profile with few or no off-target effects. Therefore, populations who are the most vulnerable to malaria, i.e., pregnant women and young children would have access to such new treatments much faster than is presently the case for new antimalarials. An analysis of mAbs that were successfully developed for oncology illustrates some of the feasibility aspects, and their potential as affordable drugs in low- and middle-income countries.

## 1. Introduction

One of the major shifts of the past decades in how medicine is practiced is a transition from treating patients—waiting for them to become sick—to preventive medicine. This process is driven by better observational data and statistics of who is at risk of disease, and the availability of better and safer medicines (Table 1). The earliest of these approaches came from the cancer field (‘chemoprevention’ is often exclusively defined as applied to this disease [1]), where patients whose primary tumor was excised are routinely treated with cytostatic drugs (and irradiation) to prevent recurrence, without knowing what patients might benefit. Later on, it became apparent that hypertension and hypercholesteremia are risk factors for cardiovascular disease, and millions of people today take drugs that revert such conditions which are, in essence, biomarkers, or ‘signs’ (medical observations), and not symptoms of disease. Similarly, hyperglycemia leading to type II diabetes is now aggressively treated with drugs, to prevent irreversible organ damage. These concepts are now so well established that new drugs are routinely approved on the basis of evidence that they can normalize these biomarkers, rather than demonstrating a benefit in preventing disease. Because of this amalgamation of biomarker and disease, the term ‘chemoprevention’ is rarely used in the last three situations, but the principle—giving drugs to prevent disease—is very similar.

Chemoprophylaxis (see Table 2 for definitions of these terms in the malaria field) is also becoming increasingly important for infectious diseases, where vaccination has an even longer and highly successful track record. An increasing number of safe and efficacious medicines to treat malaria and other infectious diseases have been developed, and some of these are now increasingly being used for the pre-emptive protection of populations. To date, such campaigns are routinely conducted for malaria and onchocerciasis (river blindness) but other diseases, such as trachoma (*Chlamydia trachomitis*), elephantiasis, schistosomiasis, soil-transmitted helminths and scabies are also being considered in trials [2,3].

What all these treatments have in common is a requirement to be safe, where safety is to be considered against the risks of omission—risks that may be very high after cancer surgery, acceptable for chronic hyperglycemia but lower for some infectious, treatable diseases, where safety is typically compared to vaccines (for other infectious diseases). In Low- and Middle-Income Countries (LMICs) the risks are also to be evaluated in the context of delayed access to specialized healthcare for those with disease symptoms, and co-morbidities (such as malnutrition, tuberculosis and HIV infection). Another consideration is that drug administration campaigns can be relatively short, covering the rainy transmission season in Sahel countries, or address only those who are at increased risk of developing severe outcomes: pregnant women and immune-naïve young children and travelers. Such campaigns may also reduce the overall pathogen density in an area and, while perhaps not eradicating a disease, result in durable relief. In the case of malaria, the focus of this review, an important risk of such campaigns is the spread of antimalarial resistance. This risk generally increases when plasma concentrations of the protective drugs wane over time, whereby large numbers of parasites may multiply in an environment with suboptimal antimalarial concentrations, which is a breeding ground for resistance [7]. For this reason the WHO (World Health Organization) recommends using different drug combinations for mass drug administration and therapy for a given area [8]. With the steady spread of resistance against antimalarials, new medicines are needed to replenish the portfolio and to continue to honor this precautionary principle.

As noted above, the cancer field has pioneered the concept of chemoprevention over half a century ago, and is now again being looked at by the malaria community for its successful discovery of therapeutic monoclonal antibodies with long (weeks) plasma half-lives that can be mass-produced at affordable prices (as far as production costs are concerned [9,10] ). This review will look at the history and types of existing chemopreventive practice for malaria and discuss the potential of long-acting biologics and small-molecule drugs for this area.

## 2. Prophylaxis for Malaria

### 2.1. The Four Vulnerable Populations: (Pregnant, Children, Travelling and Non-Immune or Malaria-Naïve Populations)

The parasites that cause malaria—five protozoan species of *Plasmodium*, transmitted by *Anopheles* mosquitoes [11]—impair their host’s immunity [12], which allows the parasites to multiply to astronomical numbers (up to around 10^12^ organisms) before entering a little-understood process leading to self-limitation that involves quorum-sensing or residual host immunity. Such a self-limiting mechanism makes sense—a well-known ecological rule says that infectious pathogens tend to keep their victims alive (become less virulent over time); exceptions occur when a disease is ‘new’ to a population (such as the 14th C Plague), when a pathogen ‘accidently’ spills over into a different host species (rabies, Ebola virus in humans), if the death of the host is part of the parasites’ natural host changing cycle (toxoplasmosis, as it cycles between rodents and cats), when death occurs only after a lengthy period of time (HIV) or involves smaller subpopulations of a species (e.g., the weak, the old—a situation that is likely to apply to SARS-CoV-2).

In the case of malaria, immunity in adults in endemic regions is only partial, and the *Plasmodium* density ‘set point’ takes this level of immune proficiency into account, so parasite densities do not cross a threshold where the risks for severe (cerebral) malaria and death increase (indeed, severe malaria is partially defined by hyperparasitaemia [13]). The problem with this situation is that building up the semi-protective immunity takes time—with years of repeated disease episodes.

The most vulnerable cohort in terms of malaria deaths are children younger than five years old. In 2018, they accounted for 67% (272,000) of all malaria deaths [4]. The second vulnerable population consists of pregnant women. Such women are at increased risk of more frequent and severe malaria than non-pregnant women [14]. For unknown reasons, pregnancy increases the risk of contracting malaria [15], and the outcomes are worse; a study in Thailand found that 60% of the pregnant women who contracted malaria ended up with the severe form of the disease [14]. As a result of these two risk factors, 5–10% of all pregnant women in Africa develop severe anemia [14]. A possible explanation is that pregnancy is associated with an altered immune response that prevents rejection of the fetus and offers a pause in some autoimmune diseases such as multiple sclerosis [16], but also results in increased susceptibility to malaria and several other infectious diseases [17]. Developing malaria during pregnancy also has detrimental effects on the fetus, resulting in fetal loss, low birth weight and delayed or impaired development of the infants [18].

A third group who is particularly susceptible to severe malaria are immune-naïve travelers that enter disease-endemic regions especially of sub-Saharan Africa from regions where malaria is not endemic. 

The fourth at-risk populations are those in regions where malaria re-emerges. One scenario where this may occur is when malaria has been eliminated from a region but then returns a generation or so later. The newly immune-naïve population will then be highly susceptible to death from complications of the disease. In fact, in the absence of disease, natural malaria-induced immunity is believed to have a half-life of only about five years [19], and the above scenario can develop quite rapidly. There are many reasons why malaria can be re-introduced into a previously malaria-free region: climate change, migration, increases in population density, a general breakdown of local healthcare (e.g., because of emerging epidemic with Ebola virus, a pandemic such as COVID-19, or war) or the loss of efficacy of long-lasting insecticidal nets (LLINs; due to the rise in mosquito resistance against pyrethroids). Because the elimination of *Anopheles* is not a realistic prospect, the return of malaria always remains a real threat. The earliest evidence for such a process—that an immune-naïve population is exposed anew to malaria—is an upward shift in the age of malaria mortality. [20,21,22,23,24,25].

### 2.2. Children

Newborn children from mothers with partial immunity against malaria enjoy some protection from maternally transmitted antibodies, but these only last for a few months, after which the babies enter a period where they are at high risk of developing severe malaria and dying, until they are about five years old. For this population, two protective interventions are defined [26]. First, intermittent preventive treatment in infants (IPTi) (Table 3), administration of SP during routine vaccination visits provided to infants up to 12 months of age in areas of moderate to high malaria transmission, and second, seasonal malaria chemoprevention (SMC; Table 3) was introduced, a form of mass drug administration (MDA) that consists of a once-monthly full course of sulfadoxine-pyrimethamine + amodiaquine (SPAQ) for the three-to-four month rainy season in the Sahel sub-region of sub-Saharan Africa, where *P. falciparum* is sensitive to both antimalarial medicines. Such programs have also been extended to countries closer to the equator with longer rainy seasons, such as Senegal [27] and Nigeria [28]. SPAQ is bitter and is often reflexively spat out by young children. In order to overcome this, a child-friendly, taste-masked dispersible formulation was recently developed by the Medicines for Malaria Venture (MMV) and Guilin Pharmaceutical, and in 2019, 96 million courses of this SMC were delivered to 13 countries (available online: www.mmv.org (accessed on 18 March 2022)).

Investigating the interactions between different interventions could lead to increased protection; for example, the combination of SPAQ with the anti-malarial vaccine RTS,S has shown encouraging results as the combination of both interventions resulted in a substantially lower incidence of uncomplicated malaria, severe malaria and death than either intervention alone [29].

### 2.3. Pregnant Women

The WHO recommends that African women in moderate-to-high malaria transmission areas receive intermittent preventive treatment in pregnancy (IPTp) in the form of sulfadoxine-pyrimethamine. Treatment is to start no earlier than the second trimester, in the form of three doses, given one month apart. Whilst a trial in Indonesia found that intermittent screening and treatment (IST) with dihydroartemisinin–piperaquine (DHA-PIP) was more effective at reducing malaria in pregnancy Table 3) [30], a recent meta-analysis of trials in Africa comparing Artimisinin-based IST (ISTp-ACT) with SP-based IPTp (IPTp-SP) showed that ISTp-ACT was not superior regarding the prevalence of malaria infection at delivery, and there was a higher risk of subpatent infections associated with low birthweight and preterm delivery with ISTp-ACT [31].

### 2.4. Travellers

Historically speaking, the chemoprophylaxis of malaria for travelers is the earliest example of this type of medicinal practice. Its origins go back to the mid-19th C, after quinine was first isolated as the active principle from the bark of the Peruvian cinchona tree (in 1820), vastly improving on the extracts that had been used previously [32]. This discovery had enormous historical (and dramatic) consequences, helping to finally trigger the colonial ‘Scramble for Africa’. Nowadays, quinine is used more rarely because of widespread resistance to the drug by *Plasmodium falciparium*, which causes the deadliest form of the disease. The most widely used medicines by travelers and other forms of chemoprophylaxis are listed in Table 3.

### 2.5. General Considerations for Chemoprophylactic Treatments

As noted earlier, the recommendation for malaria prophylaxis is to use different drugs as are used for case management, but *Plasmodium* parasites have developed resistance against all antimalarials that are currently in use [33]. Fortunately, in many cases resistance is regional. There is also evidence of fitness costs for the parasite for some forms of genetic resistance [34] and, to some extent, incompatibilities between certain resistance phenotypes [35]. A general problem in drug development is that women of child-bearing potential and young children are usually excluded from clinical trials, for ethical reasons. One workaround to provide access to children is to gradually include younger patients in trials. This approach has been employed in recent studies evaluating new antimalarial combination therapies [36,37]. Another problem is to document inadvertent use of antimalarials during pregnancy and monitor outcomes. Both are long and (for registries) imprecise activities. As a result, the safety of almost all antimalarials in first-trimester pregnancy is simply unknown and typically the oldest drugs, quinine or mefloquine, are used in this category of patients which, as monotherapies, are not universally effective. In summary, there is an urgent need for safe and efficacious antimalarials for this early phase of pregnancy [38]., where the fetus is also most vulnerable to xenobiotics. 

However, some of these assumptions are ripe for re-consideration. Monoclonal antibodies (mAbs) in particular have seen a spectacular rise in the past few years; a recent review listed 79 therapeutic mAbs that have been approved by the FDA since 1975 [39] (see also [40]), with a 2018 market value in excess of USD 115 billion. Over 500 others are in clinical development [41].

A key advantage of mAbs is that, as a drug class, they have an excellent safety record, with concerns almost exclusively target-based. Injected mAbs only start gradually being transported to the fetus near the end of the first trimester [42,43] and such treatments should thus be safe in these situations. For example, there is a European evidence-based consensus to continue treatment with anti-TNFα for inflammatory bowel disease (IBD) throughout pregnancy, given the risks of IBD also for the fetus, even with an FDA Category B/C designation for this drug [44]. In summary, an antimalarial mAb could be deployed much faster in vulnerable populations than an NCE (new chemical entity). This, and other injectable alternatives will be further discussed below.

Perpectives for injectable antimalarials.

There are different categories of antimalarials, and ‘Target Product Profiles’, or TPPs, have been defined and updated for each of these [45,46]. These include:TPP1, for the treatment of adults and children for malaria;TPP2, for chemoprotection.

These profiles were drafted with orally available medicines in mind, but TPP2 was recently extended for an injectable prophylactic drug [10]. Among the key essential requirements are that the drug can safely be used in babies over six months old as well as adults, it has a shelf life of at least two years and a cost that is comparable to SMC, that is, less than 5 USD per injection.

Both TPPs usually comprise a combination of two or more drugs that are defined by ‘Target Compound Profiles’ (TCPs): TCP1 drugs target the asexual (disease-causing) blood stage;TCP3s target liver hypnozoites (preventing relapse especially of *P. vivax*, predominantly non-African parasites that can remain dormant for weeks, months or—anecdotally—even years);TCP4s, targeting liver schizonts (the liver stage just prior to the blood stage);TCP5s, killing gametocytes (which are not pathogenic but transmit the parasites back to mosquitoes, where they mate and develop into infectious sporozoites) andTCP6s, endectocides, which are drugs that, when taken up by mosquitoes, kill the insect, or otherwise prevent gametocyte maturation [47].

TCP2 has been retired and was not re-purposed to prevent confusion with earlier literature. Note that only TCP1 in this list concerns drugs that cure malaria patients—all others are, to some extent, ‘chemopreventive’ drugs. As for TPP2, injectable formulations for a TCP-4 molecule were also defined [10]. 

As the list of TCPs shows, the life cycle of malaria parasites (Figure 1) can be targeted at several points. TCP1 drugs kill the asexual parasites but most often leave gametocytes that transmit the disease alone. The discovery of new antimalarials for the ‘other’ stages is highly dependent on the availability of good assays that both faithfully reflect the situation in vivo and are amenable to high-throughput screening. Both conditions are met for blood-stage (erythrocytic schizonts) *P. falciparum* only, which has led to the current harvest of TPP1 candidates [48]. The 8-aminoquinolines primaquine and tafenoquine, a drug that was very recently approved, are unique antimalarials that kill both *P. vivax* hypnozoites (TCP3) and *Plasmodium* gametocytes (TCP5), effecting a ‘radical cure’. Unfortunately, we are only slowly beginning to understand primaquine’s precise mode of action as a prodrug [49], hindering our efforts to discover additional NCEs for these TCPs. 

## 3. mAbs as Chemoprophylactic Agents for Malaria

### 3.1. General Considerations

As outlined above, mAbs offer the opportunity to:develop drugs with a safety profile that provides far earlier access to the most vulnerable malaria patients;allow *Plasmodium* parasites to be targeted at stages where they are few, reducing the risks for recurrence and resistance (a frequent problem with TPP1 drugs);overcome problems with setting up high-throughput phenotypic screens involving these earlier stages, or validating small-molecule targets;prevent both disease and transmission (unlike TPP1 medicines).

Moreover, the drug discovery process of mAbs is far more ‘rational’ than for xenobiotics, where late-stage, Phase III safety ‘surprises’ are notorious. An experimental mAb that has reached a clinical phase has a success rate of 17-25% for regulatory approval [53] as opposed to 5-10% for small molecules. Shifting the risk for failure to earlier phases of the drug discovery process is one of the best strategies to reduce overall development costs. Finally, the pharmacokinetic exposure and plasma half-life of mAbs occur in a relatively narrow range of variation in humans, which greatly helps estimating the therapeutic dose. For small molecules in contrast, metabolism and elimination are driven by cytochromes, which are encoded by genes that are highly polymorphic, particularly in sub-Saharan populations, resulting in variation in plasma exposure [54].

The case for reassessing therapeutic antibodies for neglected and tropical diseases was recently also made for schistosomiasis, fungal infections, dengue, tuberculosis, HIV, hepatitis B and visceral leishmaniasis [6] and snakebite envenoming [55,56,57], which was recently included in the WHO List of Neglected Tropical Diseases. Additionally, therapeutic antibodies are being marketed for rabies, respiratory syncytial, varicella-zoster, vaccinia and hepatitis B viruses [58,59]. These mostly concern antibody mixtures (sera) used for ‘passive immunization’.

The introduction of mAbs to prevent malaria in vulnerable populations would completely reverse today’s paradigm. Up to now we use the *oldest* antimalarials for this purpose—because they are known to be safe, and the medicine must differ from what is used in case management—for which the most efficacious medicines are used. With a preventive mAb regimen, we would use the *newest* treatment for these vulnerable populations instead, in an approach that would differ completely from how malarious patients are treated. This strategic, unprecedented turnaround comes with serious challenges for clinical development and regulatory pathway for this type of chemoprevention, but the possible strategies to overcome these have been outlined recently [10]. 

Gaudinski et al. [58] reported results from a recent phase I and CHMI trial of the monoclonal antibody CIS43LS, the half-life of the antibody after administration was 56 days, and volunteers in the CHMI part of the trial were protected from infection for a period of up to 36 weeks after administration of CIS43LS [60].

### 3.2. mAbs Optimization

Commercial mAbs are nearly all IgGs, and a number of approaches exist for these to extend their plasma half-lives to 20–25 days [61,62] and 56 days [60]. The simplest modification consists of Met428Leu and Asn434Ser amino acid substitutions in the (constant) IgG Fc region, to prevent the association of IgG with the FcRn receptor in the acidic environment of lysosomes, allowing the mAb to escape clearance by endocytosis. Such mutations extend the IgGs plasma half-life about threefold, as demonstrated by bevacizumab and cetuximab [62]. Other, older approaches to achieve the same goal exist; in essence, these make the IgGs bulkier by attaching other macromolecules [63], but these also increase the cost of goods and reduce the potency per weight unit. Allometric scaling of mAbs between mouse and humans indicates that systemic clearance is proportional to body weight raised to a power of 0.91 (Cl = a·BW^b^) [64]. This translates to a twofold difference in dosing (as measured by mg/kg) to achieve similar efficacies in mice and humans. This in turn suggests a required antigenic potency (EC_50_) below 100 pM (<15 ng/mL [65,66]. It will also be important to show that there is no antibody-dependent enhancement [67].

### 3.3. Costs

mAbs still suffer from the perception that they incur high costs of goods, an idea that is kept alive by the high list price for such drugs in the oncology field and other indications. However, for mAbs that are produced at high volumes in highly expressing producer cells at around 4 g/L production costs drop to around USD 100-300 per gram [68]; later estimates cite costs of USD 35-85 [69], which could even drop to 20 USD/g for multi-tonne scales [10]. Unlike cancer treatments, where the drugs are often administered intravenously, a malaria mAb would need to be injected intramuscularly, and that would set a limit on its volume. Some mAbs are soluble up to 200 mg/mL, but half of this concentration is a more realistic figure [70]. With maximal injection volumes of 0.5 and 2 mL for children and adults, respectively, this would set a ceiling for the mAb dose at 50-200 mg. Excluding the costs of packaging, storage and transport this would set the production costs for the mAb at USD 1.75 for children and USD 7 for adults. This would compare to the costs of a full vaccination of one child with the partially effective RTS,S vaccine of USD 5 (again excluding all other costs [71]). 

### 3.4. Safety

As pointed out previously, the transfer of maternal (or injected) antibodies to the fetus picks up only slowly and gradually near the end of the first trimester [43]. While host-directed mAbs are obviously a concern in other diseases, one can expect a *Plasmodium*-directed antibody to be specific. The rapid availability of such antibodies to populations who are usually excluded from clinical trials would be a tremendous advantage.

### 3.5. Efficacy and Resistance

The low parasite densities when they pass through the early development stages in humans would make them suitable targets for mAbs, however, these phases tend to be short. Thus, sporozoites that are injected by mosquitoes in the skin travel to the liver in about 30 min, and merozoites exist in free circulation for only 30 s [10]. Nevertheless, the RTS,S/ASO1 vaccine targeting a sporozoite surface protein [72] and sporozoite-based PfSPZ vaccines show efficacy in preclinical and human volunteer models [73,74,75], suggesting that at least the sporozoite window is not too short.

Resistance against any new therapeutic is always a possibility. However, given the small numbers of parasites at these stages, and their limited proliferation the risks appear small in this case. During clinical evaluation of the mABs in phase II and III studies and also after launch, breakthrough infections need to be monitored for mutations that leading to reduced mABs recognition. A combination of mABs or mAB and small molecule LAI could be envisaged to prevent the emergence of resistance. 

### 3.6. Potential Antigens

Because of their lower abundance, the pre-erythrocytic stages of *Plasmodium* have not been extensively studied for specific extracellular markers. A publication in 2000 identified a number of stage-specific *P. falciparum* transcripts [76], but only a subset of these have been explored as potential antigens (Table 4). A mAb that targets an antigen that is expressed both in early and blood stages may have prophylactic activity, but there is also a concern that individuals may have been tolerized.

#### 3.6.1. Targeting the Sporozoite

CSP-1 (circumsporozoite protein; see Table 4 for abbreviations) is represented by its NANP repeats in the RTS,S/ASO1 vaccine [72] and mAbs generated from vaccinated individuals prevent malaria infection in mice [77,78,79]. Protective Abs against CSP-1 are also induced naturally [80]. Several CSP-1 Abs have been characterized[78,80,81]: (see below). Ref. [82] described mAbs with potent activity both in vitro and in vivo. Recently, a human mAb (CIS43) against CSP-1 was identified [81]. It was derived from volunteers immunized with attenuated sporozoites, and active in mice. Interestingly, CIS43 prevents proteolytic processing of CSP. The authors suggest that such a mAb could be optimized for higher potency and longer half-life. A different group demonstrated further that in vivo maturation of antibodies against the CSP-1 NANP repeat region also involves selection for homotypic interactions (i.e., antibodies laterally interacting while they bind the repeat [83]). At doses of 10, 15 or 7.5 mg/kg, today’s most potent CSP-1 mAbs effect a (^10^log) fold reduction in parasite loads of 2.2 [75], 2.5 [76] and 4.9 [79] in vivo, which begins (for the latter) to reach efficacies seen for DSM265 (a small molecule inhibiting *P. falciparum* dihydroorotate dehydrogenase)) in humans [84,85]. An overview of in vitro and in vivo inhibition of malaria infection by mABS against CSP was presented by Livingston and colleagues [86].

Another sporozoite antigen candidate is CelTOS. Recently, two mAbs were described, however, these were only partially or not at all active in a mouse model [89]. Earlier, ref. [90] had analyzed sterilizing antibodies produced in mice. 

For the prevention of *P. vivax*, the binding of the parasite to the human Duffy protein can be exploited, Rawlison and colleagues have demonstrated that mABs binding the *P. vivax* Duffy binding protein has also been shown to inhibit *P. vivax* invasion [91].

The efficacy of combining different mABs directed against separate epitopes of the CSP-1 has not shown any additivity or even synergy. The combination of different mABS against repeat region and C-terminal domain of CSP-1 provided no additional protection against *P. falciparum* infection whilst repeat region mABs conferred increased protection when administered in combination with R21 vaccine induced polyclonal antibodies [92].

#### 3.6.2. Targeting the Liver Stages

There are relatively few investigations into mAbs that address the liver stage [88],. Liver-Stage Antigen 1 (LSA-1) was immunogenic in monkeys, but Ab efficacy was not evaluated [93]. Other antigens such as LSA-3, CelTOS (Cell-traversal protein for ookinetes and sporozoites) and EXP-1 (Exported protein-1) have been explored as antigens, but no antibodies have been described.

#### 3.6.3. Targeting Gametocytes

Pfs25 is uniquely expressed on female *Plasmodium* gametocytes. Pfs25 antigen sites have been compared for optimal Ab blocking [94]. Several workers [95,96,97] tested anti-Pfs25 sera in Standard Membrane Feeding Assays (SFMAs). These sera show activity in such assays but were not further characterized (no mAbs have been described), as the focus is on vaccine development. Vaccination with this antigen generates potent Abs [98].

Recently, it was shown that Abs that target Pfs47, an antigen on gametocytes of both genders, also block transmission; however, again this was focused on vaccine development [99]. 

#### 3.6.4. Targeting the Host

Apart from the parasite, there have been demonstrations that mAbs that target host proteins can halt infection, for instance, targeting basing on erythrocytes [100]. However, targeting host proteins to achieve prophylaxis could compromise safety, and the preclinical and clinical development strategy will be complicated as host interactions have to be extensively studied.

## 4. Small Molecule Drugs as Long-Acting Injectables

### 4.1. Existing Examples of Long-Acting, Injectable Drugs

Probably the oldest example here is benzathine penicillin G, which has been injected for syphilis since the 1950s [101]. More recent examples of successfully developed long-acting injectable (LAI) formulations involve antipsychotics [102]. Two first-generation drugs with this indication, fluphenazine and haloperidol, have been derivatized as decanoates. These worked well but had considerable compliance problems. Three other antipsychotics that were re-formulated as LAIs are however still in use:RLAI is a long-acting intramuscular (*i.m.)* injectable formulation of 25–50 mg risperidone, a microsphere preparation with a release duration of about 14–21 days.Paliperidone palmitate is a long-acting *i.m.* injectable formulation with 25-150 mg doses.Olanzapine pamoate is another crystal salt that is injected *i.m.* with 200–400 mg doses.Aripiprazole is simply injected monthly as a 400 mg depot *i.m.*, but this is still an experimental procedure.

HIV therapy and pre-exposure prophylaxis (PrEP) is also being explored for such applications. The example of 200–800 nm rilpivirine nanoparticles was reviewed [103], prepared by wet milling, a formulation that can be administered *s.c*. Another long-acting parenteral example is a myristoylated cabotegravir prodrug [104], formulated as crystal nanoparticles for *i.m.* use. Nevirapine was formulated as large (>50 μm), monodisperse particles coated with biocompatible polymers [105]. One of the main challenges for all these drugs is to achieve adequate plasma exposure [106]. LAI cabotegravir is presently the frontrunner in this area. When packaged into nanoparticles (GSK744LAP) the drug results in an exceptionally long plasma half-life of 21–50 days following a single dose [107].

The many approaches listed in these examples (decaonates, microspheres, pamoates, palmitates, wet milling, nanoparticles, polymer coating or simply bulk injections) illustrate that the chemical nature of the drug and its pharmacodynamic behavior are the primary determinants of what formulation will work.

Dosing regimens of these interventions, are once every 3 months achievable.

### 4.2. Reformulating Antimalarials as Long-Acting Injectables

Traditionally, drug discovery for malaria has focused on orally available medicines for reasons of cost, the ease of transport and storage, and convenience. Efforts are underway to develop current oral antimalarials as LAI; with its partners, MMV has started to assess i.m. formulations of atovaquone, proguanil, P218, piperaquine and pyronaridine to provide a protective cover of at least 3 months ([108] and mmv.org). 

### 4.3. An Outline of a Long-Acting, Chemopreventive Antimalarial

The widest use of a LAI antimalarial—a NCE or an mAb—would be as an alternative for SMC. A well-designed mAb or small molecule formulation might provide protection throughout an entire rainy season of 3–4 months from a single injection. The simplest NCE formulation approaches would use an oil solution or suspension of a micro- or nano-particulate drug, since these are low-cost and allow heat sterilization. Suitable excipients are fractionated coconut, castor, corn, cottonseed and olive oils, which have good local tolerability [109]. Milling the drug substance to the desired particle size is not trivial, as is for example attested by the variable quality of liposomal formulations of amphotericin B [110]; as used for antifungal treatments). An extended drug release period also adds to the complexity of pharmacokinetic considerations. This type of drug could be administered s.c. or i.m. The former approach is easier, but only allows smaller volumes. As pointed out earlier, i.m. injections allow maximal injection volumes of 0.5 and 2 mL for children and adults, with a suggested 27-gauge needle size to limit discomfort. These constraints are no absolutes; microcrystalline penicillin G is routinely given in 4 mL volumes, administered using a 21-gauge needle [111], even if the procedure is painful. In light of such constraints, a single treatment per season is essential.

## 5. Discussion

The number of malaria deaths have seen a steady decline since the turn of the Millennium [4] Eleven countries have been declared malaria-free by the WHO during the past two decades (Algeria, Argentina, Uzbekistan, Paraguay, Kyrgyzstan, Sri Lanka, Maldives, Armenia, Morocco, Turkmenistan and United Arab Emirates). In the year 2000, 106 countries had ongoing transmissions of malaria but 57 reduced malaria incidence by more than 75 percent by 2015 [112]. These successes have been driven by the widespread distribution of long-lasting insecticidal nets (LLINs) and the introduction of ACTs that are used both in case management and SMC. However, this is not a time for complacency, as each of the weapons that allowed us to win these battles are under a serious threat of obsolescence. LLINs rely on a single class of insecticides (pyrethroids). The resistance of *Anopheles* mosquitoes against this class is rapidly rising [113], and few other insecticides are suitable for producing LLINs. The deployment of LLINs and other antimalarial measures are highly dependent on external funding and ease of deployment, and both are under severe threat by the COVID-19 pandemic. As noted earlier, resistance against the latest class of antimalarials, the artemisinins, is becoming more frequent in SE Asia, and its spread to Africa has been reported in Rwanda [114] and Uganda [115].

History teaches us that successes against malaria may not last, such as those affected by the DDT campaigns in the 1960s and the near-elimination and rebound of malaria in Sri Lanka [116,117]. Until malaria is truly eradicated globally, all tropical regions remain at high risk.

Both treatment and chemoprevention of malaria have traditionally relied on the steady discovery of new chemical classes, recently boosted by the development of high-throughput screening assays for blood-stage *P. falciparum*, and the access to large chemical libraries [48]. However, this model is not sustainable enough to out-run the emergence of resistance. LAI antimalarials have an important role to play in making the transition to medicines—small molecules and mAbs—that act at the earliest stages of the parasitic cycle. This requires the further development of assays that reflect such stages to test such molecules in, preferably in a high-throughput setting. It also requires new ways of running clinical trials, and their regulatory pathway. Such molecules must be as safe as vaccines, further complicating how they are to be tested. However, overcoming these challenges will bring multiple benefits. Such medicines not only prevent disease but also the generation of gametocytes, thus breaking the transmission cycle, a major benefit to the community as a whole, and a significant ethical consideration. Any risk of a preventive treatment should therefore not only be measured against the benefit of remaining malaria-free during the period that the drug is active, but also beyond, since the chance that the parasite cycles back is reduced as well. Importantly, the most vulnerable populations—children and pregnant women—would get far earlier access to such medicines, especially mAbs, upon approval. Additionally, such treatments easily comply with the WHO recommendation that prevention and treatment should use different drug combinations. The present reduction in global malaria cases provides a unique but transient time window to move into this direction.

The combination of different approaches to prevent the transmission of malaria has shown the most significant and lasting effects, therefore, the use of LAI in the prevention of malaria infections should not be used in isolation and should always be employed in combination with nonmedical prevention approaches to ensure a lasting effect such as in elimination or eradiation settings.

The development of therapeutic antibodies against SARS-CoV-2 infections highlighted the potential speed from concept to clinic to registration for mAbs-based interventions [118]. Whether this could be transferred to neglected and poverty-associated diseases such as malaria remains to be seen. There are many hurdles, for instance, the interest of pharmaceutical companies to invest in these diseases is limited, development of a mAb against a neglected disease would divert scarce resources especially manufacturing capacities from profitable indications. Another hurdle to the implementation of an injectable prevention is the targeted treatment duration, which will require substantial quantities of either a small molecule or mAb. Therefore, a pediatric focus might be the principal focus. 

## Figures and Tables

**Figure 1 tropicalmed-07-00058-f001:**
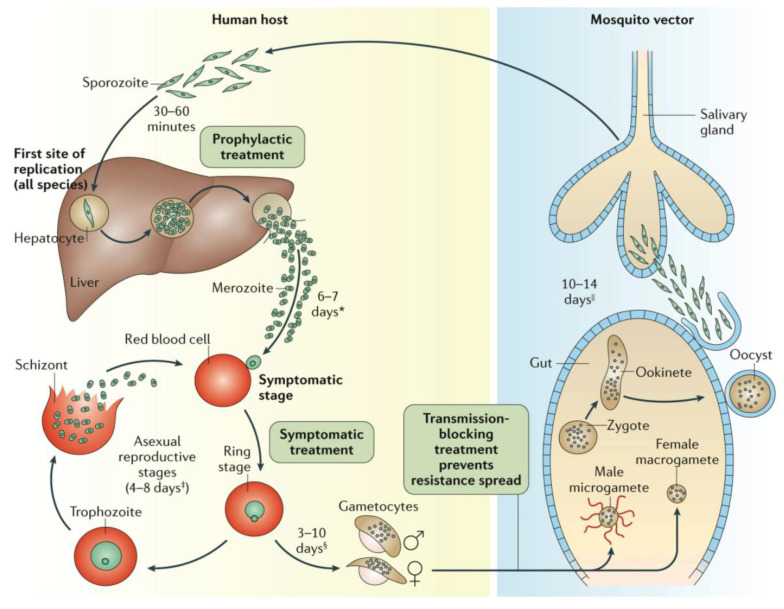
The *Plasmodium* spp. life cycle Reprinted with permission from ref. [11], Copyright 2017 Springer Nature Targeting the earliest *Plasmodium* stages (sporozoites, liver schizonts and hypnozoites) is attractive because parasite numbers at this stage are very low, and a successful drug will not only prevent disease but also production of the infective gametocytes. It is unlikely that a single mAb can replace a TPP1 medicine. For instance, an observation of recrudescent vs. cured malaria patients suggested that host responses that generate antibodies against EXP1, MSP3, GLURP, RAMA, SEA and EBA181 contribute towards eliminating the parasite from patients [50,51,52]. However, these Abs only prevented in vitro infection of erythrocytes when combined [51]. mAbs have, however, excellent potential for all other TCPs. Here, the bottlenecks are the availability of good assays, and suitable antigenic targets. As pointed out earlier there are few good, ‘screenable’ assays for the important early biological processes, however, there is increasing molecular insight in surface markers of these stages and their roles, and such markers can be targeted by mAbs, not only to interfere with the activity of these *Plasmodium* stages but also to target them for destruction by the immune system. * Merozoite invasion of red blood cells can be delayed by months or years in case of hypnozoites. ^‡^ The number of days until symptoms are evident. ^§^ The duration of gametogenesis differs by species. ^||^ The maturation of sporozoites in the gut of the mosquito is highly temperature-dependent.

**Table 1 tropicalmed-07-00058-t001:** Standard ‘chemopreventive’ treatments today, as defined by the use of medicines to prevent disease (and excluding vaccination).

	Disease That Is to Be Prevented	Biomarker, Condition	Drug Administration
Non-infectious disease	Coronary artery disease, heart failure	Hypercholesteremia	Low-density lipoprotein-lowering drugs, e.g., statins
	Cardiovascular disease	Hypertension	Anti-hypertensives, e.g., angiotensin-converting-enzyme inhibitors
	Diabetes (type II)	Fasting-state hyperglycemia	Antidiabetics, e.g., metformin
	Cancer	Resection of primary cancer with risk for recurrence	Chemotherapy or chemoprevention (cytostatic drugs)
Infectious disease	Malaria	Healthy individuals at risk for infection	Mass drug administration, seasonal malaria chemoprevention, intermittent protection in pregnancy or children and chemoprophylaxis in travelers with antimalarials
	River blindness (*Onchocerca volvulus* infection)	Healthy individuals at risk for infection	Mass drug administration with ivermectin

**Table 2 tropicalmed-07-00058-t002:** Definitions of preventive medications for malaria [4,5,6] for the last term.

Term	Definition
chemoprophylaxis	Generic term for treatments aimed at preventing malaria-in travelers and other non-immunes exposed to malaria transmission
chemoprevention	The administration of full curative treatment courses, typically administered during seasonal chemoprevention (SMC)or intermittent treatment in pregnancy (IPTp) and infants (IPTi)
Mass Drug Administration	The administration to all age groups of a defined population (except those for whom the drugs are contraindicated) at the same time regardless of infection status, to accelerate malaria elimination through rapid and sustained reduction of transmission and to reduce mortality and morbidity in emergency situations
preventive therapy	Umbrella term for chemoprophylaxis, intermittent preventive treatment of infants and pregnant women, seasonal malaria chemoprevention and mass drug administration

**Table 3 tropicalmed-07-00058-t003:** Chemoprophylactic regimens for malaria. ^1^ Treatments recommended by the Centers for Disease Control and Prevention (CDC); ^2^ Previously (before the 2012 WHO recommendation) named Intermittent Preventive Treatment in children, or IPTc; ^3^ Recommended by the WHO; ^4^ Recommended by the WHO; ^4^ Experimental.

Population	Type of Chemoprophylaxis	Treatments
Children under 5 years of age	seasonal malaria chemoprevention (SMC)	sulfadoxine-pyrimethamine + amodiaquine ^2^
	Intermittent preventive treatment in infants	sulfadoxine-pyrimethamine ^3^
Pregnant women	intermittent preventive treatment in pregnancy	sulfadoxine-pyrimethamine ^3^
	intermittent screening and treatment	dihydroartemisinin–piperaquine ^4^
Travelers	chemoprophylaxis	atovaquone/proguanil (Malarone); chloroquine; doxycycline; mefloquine; primaquine; Tafenoquine ^1^

**Table 4 tropicalmed-07-00058-t004:** *Plasmodium* antigens that have been explored as vaccines/antigens, by parasitic stage Adapted with permission from ref. [87] Copyright 2007 Bentham Science Publishers and [88] Copyright 2016 Intech OpenAbbreviations: CSP, circumsporozoite protein; TRAP, thrombospondin-related adhesive protein; ME-TRAP, multiple epitope-thrombospondin-related adhesive protein; STARP, sporozoite threonine- and asparagine-rich protein; SALSA, sporozoite- and liver-stage antigen; SSP, sporozoite surface protein; SPf66, synthetic P. falciparum 66; MSP, merozoite surface protein; RESA, ring-infected erythrocyte surface antigen; GLURP, glutamine-rich protein; AMA, apical membrane antigen; SERA, serine-repeat antigen; EBA, erythrocyte-binding antigen; HRH5 reticulocyte-binding protein homologue 5, EMP, erythrocyte membrane protein; RAP, rhoptry-associated protein; LSA-1/3, Liver-stage antigen-1/3; CelTOS, Cell-traversal protein for ookinetes and sporozoites; DBP, Duffy binding protein, EXP-1, Exported protein-1; Ripr, RH5-interacting protein; CyPRA, Cysteine-rich protective antigen; Pf, *Plasmodium falciparum* protein; Pv, *Plasmodium vivax* protein.

Sporozoites	Liver Stages	Blood Stages	Gametocytes
CSP-1	LSA-1	SERA	P125
TRAP	CelTOS	EBA-175	P1230
STARP	EXP-1	AMA-1	Pfg27
SALSA	LSA-3	RAP-2	Pfs45/48
SP-2	STARP	RAP-1	Pvs28
CelTOS	TRAP	GLURP	Pvs25
DPB		MSP-1	Pfs16
		RESA	Pfs28
		MSP-2	
		EMP-1	
		MSP-3	
		Pd35	
		MSP-5	
		P155	
		hRH5	
		Ripr	
		CyRPA

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
