# Peer review of "Development of New Strategies for Malaria Chemoprophylaxis: From Monoclonal Antibodies to Long-Acting Injectable Drugs"

_tropicalmed, 2022, doi:10.3390/tropicalmed7040058_

Round 1

Reviewer 1 Report

I thought I could review this article on all these axes, but in reality my reading can only be limited to the epidemiological level. The article seems to me well written and fair in its scientific content.

However, I am not knowledgeable on certain pharmacological and therapeutic issues and would recommend review by a pharmacologist before publication.

Best regards

Author Response

Editorial corrections as suggested by reviewer 1 in the pdf have been accepted

Reviewer 2 Report

  1. Use of MAbs as prophylactic interventions is not a very practical one especially in economically challenged societies. The costs for such products are prohibitive in poorer sectors of the globe. Authors have discussed this point by suggesting that the chemoprophylaxis measures using MAbs and LAIs should be contemplated for the four vulnerable populations: pregnant women, young children, travelers and relapsing populations. Some cost effectiveness discussion on this point will be very useful.
  2. The public health aspect that involves social determinants of health such as limiting mosquito breeding grounds near homes and in homes has not been given much attention.
  3. This manuscript would be better if it at least brings out the community level prevention approaches. As for most infectious disease transmission drugs are to be combined with non-medical prevention approaches to have a permanent effect such as in elimination or eradication strategies.
  4. Unclear why chemoprophylaxis for gametocytes is included. It seems a little impractical to include a discussion of MAbs for Pfs25 and Pfs47 - gametocyte antigens that appear inside the mosquito, because administering expensive agents to humans in the hopes that these agents will be taken up by mosquitoes in a blood meal and prevent parasite development is an expensive proposition. Authors should rethink the practicalities of the suggested applications regarding chemoprophylaxis.
  5. It is also a great possibility that immune pressure can lead to escape from mAbs bu that has not been discussed at all. Development of resistance to drugs can extend to mABs and LAIs can have the same problems.
  6. Minor comments:  MAbs used in some cases and mABs in other places. Need to use one abbreviation.
  7. Overall, this is a clearly written and organized report that discusses the use of mAbs and LAIs for malaria chemoprophylaxis.  Analogy is made to other infectious diseases where LAIs are being success fully used. However, it was not very clear what the take home information was from the manuscript. There was no clear action items delineated but several possibilities were suggested with no in depth discussion of the practical issues.  It will be a much better review or report if practical applications were considered and discussed in depth.

Author Response

1 Use of MAbs as prophylactic interventions is not a very practical one especially in economically challenged societies. The costs for such products are prohibitive in poorer sectors of the globe. Authors have discussed this point by suggesting that the chemoprophylaxis measures using MAbs and LAIs should be contemplated for the four vulnerable populations: pregnant women, young children, travelers and relapsing populations. Some cost effectiveness discussion on this point will be very useful.

Cost effectiveness is difficult to estimate, as discussed by Macintyre et al 2018, the TPP for a LAI intervention should be less than 5 $ for a malaria transmission season, this has been discussed in the section “Costs” of the manuscript

  1. The public health aspect that involves social determinants of health such as limiting mosquito breeding grounds near homes and in homes has not been given much attention.

The manuscript focussed on Long acting injectable intervention, limiting mosquito breeding grounds, the use of LLIN etc were not be part of the scope, I included the need to combine all measures to reduce transmission.

  1. This manuscript would be better if it at least brings out the community level prevention approaches. As for most infectious disease transmission drugs are to be combined with nonmedical prevention approaches to have a permanent effect such as in elimination or eradiation strategies

This point is now included in the discussion

Unclear why chemoprophylaxis for gametocytes is included. It seems a little impractical to include a discussion of MAbs for Pfs25 and Pfs47 - gametocyte antigens that appear inside the mosquito, because administering expensive agents to humans in the hopes that these agents will be taken up by mosquitoes in a blood meal and prevent parasite development is an expensive proposition. Authors should rethink the practicalities of the suggested applications regarding chemoprophylaxis.

Currently we do not consider mAB against gametocytes as an intervention for the reasons the reviewer stated. We have included the candidates in the manuscript as others are working on it for completeness sake

  1. It is also a great possibility that immune pressure can lead to escape from mAbs bu that has not been discussed at all. Development of resistance to drugs can extend to mABs and LAIs can have the same problems.

This point is now addressed in the manuscript

  1. Minor comments: MAbs used in some cases and mABs in other places. Need to use one abbreviation.

In the text of the manuscript mABs is consistently used, MAbs is used in the reference section when this is was used there by the authors or the journals

  1. Overall, this is a clearly written and organized report that discusses the use of mAbs and LAIs for malaria chemoprophylaxis. Analogy is made to other infectious diseases where LAIs are being success fully used. However, it was not very clear what the take home information was from the manuscript. There was no clear action items delineated but several possibilities were suggested with no in depth discussion of the practical issues. It will be a much better review or report if practical applications were considered and discussed in depth.

The scope of the review is to highlight the options for long acting injectables for the use in malaria prophylaxis, we have addressed the challenges with respect to cost of goods and delivery and I feel that a detailed and in-depth discussion of the practical applications would warrant a separate review.

Reviewer 3 Report

The review by Moehrle J.J. entitled “Development of new strategies for malaria chemoprophylaxis: From monoclonal antibodies to long-acting injectable drugs” proposes to discuss and summarize the scenario of malaria treatment and the use of mAbs as a promising strategy compared to the small molecule’s compounds especially for risk populations.

The topic is interesting for drug development, vaccine, and policies for malaria treatment. The text is well-written with good topics structure and can be a good complement to the available literature.

Minor queries

A suggestion for easy access to readers is the inclusion of a customized malaria life cycle with the indication of mAbs recognition targets mentioned in the text.

Another suggestion is the inclusion of important references uncited: Livingstone, M.C. et al. Sci Reports 2021; Alanine D.G.W. et al. Cell 2019; Rawlinson, T.A. et al. Nat. Microbiol 2019; Roth A. et al Nat Commun 2018; Wang L.T. et al Plos Pathog. 2021.

Lines 354 and 359. Citations missing.

Line 624 – ref 56. missing volume and pages.

Author Response

The review by Moehrle J.J. entitled “Development of new strategies for malaria chemoprophylaxis: From monoclonal antibodies to long-acting injectable drugs” proposes to discuss and summarize the scenario of malaria treatment and the use of mAbs as a promising strategy compared to the small molecule’s compounds especially for risk populations.   

The topic is interesting for drug development, vaccine, and policies for malaria treatment. The text is well-written with good topics structure and can be a good complement to the available literature.  

Minor queries

A suggestion for easy access to readers is the inclusion of a customized malaria life cycle with the indication of mAbs recognition targets mentioned in the text.

 Thank you for the suggestion the parasite life cycle is now included as fig 1

Another suggestion is the inclusion of important references uncited::

 Livingstone, M.C. et al. Sci Reports 2021; this reference is now included

Alanine D.G.W. et al. Cell 2019;  this reference is now included

Rawlinson, T.A. et al. Nat.Microbiol 2019; this reference is now included

 Roth A. et al Nat Commun 2018; the paper by Roth and colleagues is describing a in vitro technology for screening molecules for their liver stage activity. The discovery tools for the identification of new liver stage small molecules is outside the scope of this review and the reference was therefore not included.

Wang L.T. et al Plos Pathog. 2021. this reference is now included

Lines 354 and 359. Citations missing. I assume that this was a formatting error when the manuscript was converted to .pdf, it has been corrected

Line 624 – ref 56. missing volume and pages. Volume and pages added